# Genetic Landscape of Congenital Cataracts in a Swiss Cohort: Addressing Diagnostic Oversights in Nance–Horan Syndrome

**DOI:** 10.3390/biomedicines13081883

**Published:** 2025-08-02

**Authors:** Flora Delas, Jiradet Gloggnitzer, Alessandro Maspoli, Lisa Kurmann, Beatrice E. Frueh, Ivanka Dacheva, Darius Hildebrand, Wolfgang Berger, Christina Gerth-Kahlert

**Affiliations:** 1Institute of Medical Molecular Genetics, University of Zurich, 8952 Schlieren, Switzerland; 2Department of Ophthalmology, University Hospital of Zurich, 8091 Zurich, Switzerland; 3Department of Ophthalmology, University Clinics Inselspital Bern, 3010 Bern, Switzerland; 4Department of Ophthalmology, Cantonal Hospital of St. Gallen, 9007 St. Gallen, Switzerland; 5Neuroscience Center Zürich (ZNZ), University of Zurich and ETH Zurich, 8006 Zurich, Switzerland; 6Zurich Center for Integrative Human Physiology (ZIHP), University of Zurich, 8006 Zurich, Switzerland

**Keywords:** congenital cataract, whole-exome sequencing, NGS, Nance–Horan syndrome, *NHS*, *GJA8*

## Abstract

Congenital cataracts (CCs) are a leading cause of preventable childhood blindness, with genetic factors playing a crucial role in their etiology. Nance–Horan syndrome (NHS) is a rare X-linked dominant disorder associated with CCs but is often underdiagnosed due to variable expressivity, particularly in female carriers. **Objective**: This study aimed to explore the genetic landscape of CCs in a Swiss cohort, focusing on two novel *NHS* and one novel *GJA8* variants and their phenotypic presentation. **Methods**: Whole-exome sequencing (WES) was conducted on 20 unrelated Swiss families diagnosed with CCs. Variants were analyzed for pathogenicity using genetic databases, and segregation analysis was performed. Clinical data, including cataract phenotype and associated systemic anomalies, were assessed to establish genotype–phenotype correlations. **Results**: Potentially pathogenic DNA sequence variants were identified in 10 families, including three novel variants, one in *GJA8* (c.584T>C) and two *NHS* variants (c.250_252insA and c.484del). Additional previously reported variants were detected in *CRYBA1*, *CRYGC*, *CRYAA*, *MIP*, *EPHA2*, and *MAF*, reflecting genetic heterogeneity in the cohort. Notably, *NHS* variants displayed significant phenotypic variability, suggesting dose-dependent effects and X-chromosome inactivation in female carriers. **Conclusions**: NHS remains underdiagnosed due to its variable expressivity and the late manifestation of systemic features, often leading to misclassification as isolated CC. This study highlights the importance of genetic testing in unexplained CC cases to improve early detection of syndromic forms. The identification of novel *NHS* and *GJA8* variants provides new insights into the genetic complexity of CCs, emphasizing the need for further research on genotype–phenotype correlations.

## 1. Introduction

Congenital cataract (CC), defined as any opacity in the crystalline lens evident at birth, remains a significant contributor to treatable childhood blindness on a global scale, affecting approximately 1 to 10 per 10,000 live births [1,2,3,4,5]. Nearly half of all CC cases are attributed to genetic factors, with inherited forms displaying considerable phenotypic diversity [5,6]. CCs are primarily inherited through an autosomal dominant (ad) pattern; autosomal recessive (ar) and X-linked recessive (xlr) patterns occur less frequently [7]. To date, *NHS* is the only gene known to cause CCs through a clearly defined X-linked dominant (xld) inheritance pattern in males and females [7,8]. Two other X-linked genes, oculocerebrorenal syndrome of Lowe (OCRL) and oculofaciocardiodental syndrome (OFCD), can also be associated with isolated/syndromic CCs but follow different inheritance patterns: OCRL is X-linked recessive, while OFCD is male-lethal X-linked dominant [9,10]. Vision loss is often due to amblyopia but can also result from complications like secondary (aphakic) glaucoma or retinal detachment after cataract surgery [11]. A total of 70% of CCs occur in isolated form, while 15% co-occur with other ocular anomalies (i.e., microphthalmia, aniridia, anterior segment dysgenesis (ASD), and/or retinal anomalies). The remaining 15% are linked to broader syndromic or metabolic conditions, such as trisomy 21, galactosemia, OFCD syndrome, and OCRL syndrome [3,6,12,13,14,15]. One other syndromic form is Nance–Horan syndrome (NHS), a rare X-linked dominant disorder characterized by nuclear CCs in male patients, though other ocular anomalies like microcornea and microphthalmia are frequently reported, as well as the systemic manifestations with dental anomalies and facial dysmorphisms [16,17,18]. Dental features most commonly manifest with screwdriver-shaped incisors, supernumerary teeth, interdental gaps, and pointed premolar/molar cusps. Facial features include a long, narrow face, bulbous nose, protruding ears, and, in some cases, intellectual disability is documented [18,19]. The exact prevalence of NHS remains undetermined [20]. Despite its distinctive features, diagnosing NHS can be challenging. CCs often serve as the earliest manifestation but may not be interpreted as a syndromic sign, especially in female patients where lens opacity tends to be less severe [18,21]. CCs may also be unilateral and therefore not necessarily prompting genetic workup. Notably, NHS characteristics such as dental anomalies or intellectual disabilities may not become apparent until later in development or may be completely absent, further hindering timely diagnosis [18].

CCs are classified by localization (polar, nuclear, lamellar, cortical), opacity type (solid, pulverulent, crystalline, blue dot), and sutural opacity presence. Subtypes include nuclear, total, pulverulent, cerulean, and polymorphic, aiding precise diagnosis and management [5]. In males affected by NHS severe dense CCs typically involve the nucleus and posterior Y suture and may extend into the posterior cortex. In contrast, females with NHS usually exhibit milder lens opacities, often manifesting as posterior Y-sutural cataracts that generally do not significantly impair vision in childhood [19,21].

CCs exhibit significant genetic heterogeneity, involving both Mendelian and non-Mendelian mechanisms. The Cat-Map database, a curated resource for cataract-associated genes and loci, has cataloged over 450 genes to cataracts, including more than 300 genes associated with syndromic forms, reflecting the increasing complexity of genetic contributions to cataractogenesis [3,22,23]. Non-Mendelian cataracts include those resulting from mitochondrial inheritance, such as in Sengers syndrome—a rare autosomal recessive mitochondrial disorder characterized by CCs, hypertrophic cardiomyopathy, and lactic acidosis due to variants in the *AGK* gene [24,25]. Additionally, epigenetic modifications, like oxidative stress-induced change in DNA methylation, can suppress antioxidant defense mechanisms, contributing to lens opacification [26,27]. Nearly half of these mutations are found in crystallin genes, approximately 25% affect membrane protein genes, such as connexins and major intrinsic proteins, and the remaining genes include growth regulators, transcription, and the cytoskeleton [6,28,29]. Additionally, more than 10 loci linked to inherited cataracts have been mapped to specific chromosome regions, but the exact genes involved remain unidentified, making these “orphan” loci [8,22]. For *NHS* specifically, 102 disease-causing variants have been reported to date according to the Human Gene Mutation Database (HGMD) (https://www.hgmd.cf.ac.uk, accessed on 6 May 2025). Compared to other genes implicated in CCs, *NHS* variants represent a very small subset, underscoring their rarity in the broader genetic landscape. This extensive genetic and phenotypic heterogeneity poses challenges for establishing precise genotype–phenotype correlations for CCs, which are crucial for accurate diagnosis and genetic counseling [12,30].

Advancements in next-generation sequencing (NGS) have revolutionized gene discovery for CCs by enabling high-throughput analysis through methods like targeted panels, whole-exome sequencing (WES), and whole-genome sequencing (WGS) replacing the slower and less comprehensive traditional Sanger sequencing technology [22,31]. Among these methods, WES, in particular, offers a higher diagnostic yield by targeting protein-coding regions: the regions most likely to harbor pathogenic variants associated with CC [30,32,33]. For these reasons, we selected WES for this study to ensure a robust and comprehensive approach to identifying disease-causing variants in a Swiss cohort of patients with CCs.

In our study, we identified potentially pathogenic genetic variants in 10 families through WES. Among these, we discovered three novel variants: *GJA8* (Gap junction alpha) (c.584T>C) and two distinct *NHS* variants, *NHS* (c.250_252insA) and *NHS* (c.484del). These findings emphasize the genetic heterogeneity of CCs and provide new insights into genotype–phenotype correlations. Notably, *NHS* variants demonstrated phenotypic variability consistent with dose dependency, meaning that clinical severity may relate to gene dosage in X-linked dominant inheritance. While hemizygous males often exhibit most syndromic features, including CCs, expression is variable. Heterozygous females typically show no, milder, or isolated findings. This pattern, also observed in prior studies [21], underscores the role of gene dosage and potential modifiers in *NHS*-related cataract expression.

## 2. Materials and Methods

### 2.1. Study Subjects

This study was conducted as part of the cataract genetics study led by the Department of Ophthalmology at University Hospital Zurich, in collaboration with the Institute of Medical Molecular Genetics at the University of Zurich. The research aimed to achieve detailed phenotypic and genotypic characterization of CCs. The cohort included 20 unrelated probands from 20 families, all of whom had CCs without a prior genetic diagnosis and provided written informed consent. Subjects were recruited through partnerships with ophthalmic centers across Switzerland, including the University Hospital Zurich, University Clinics Inselspital Bern, and Cantonal Hospital St. Gallen. All subjects underwent regular comprehensive eye examinations upon CC diagnosis and a retrospective chart review was conducted gathering the medical history. Detailed pedigrees, reporting affected family members, were collected to support genetic analysis. Information on the type of cataract at the initial presentation prior to surgery, associated ocular and systemic anomalies, surgical timing of cataract surgery, and postoperative outcomes, sequelae and their treatment, habitual visual acuity (VA) and age were extracted from the medical records of genetically solved subjects. Inclusion criteria were patients with binocular CCs, independently of the status of cataract surgery, unknown molecular genetic cause, consent to participate in this study, and access to patient charts. Blood samples were obtained from the subjects as well as both of their parents for genetic analysis if available. This study adhered to Good Clinical Practice standards and the principles outlined in the Declaration of Helsinki [34]. Ethical approval was granted by the Cantonal Ethics Committee of Zurich (Ref-No. 2019-00108), and written informed consent was secured from all participants or their legal guardians in the case of minors. No compensation or incentives were offered to participants for their involvement in this study.

### 2.2. Target Genes

The gene list (Appendix A) utilized in this study is consistent with the one compiled by Delas et al. (2023) [6], which builds upon the earlier work of Rechsteiner et al. (2021) [30]. The list was augmented by incorporating data from the Human Gene Mutation Database (HGMD) and conducting an extensive review of recent literature. The list includes cataract-associated candidate genes, covering both syndromic and non-syndromic phenotypes, as well as cataract-related genes identified in animal models.

### 2.3. Exome Sequencing and Data Analysis

We performed WES as described previously [6,28,33,34]. DNA was extracted from venous blood using the Chemagic DNA Blood Kit (Perkin Elmer, Waltham, MA, USA), and fragmented with the M220 Sonicator (Covaris, Woburn, MA, USA). WES libraries were prepared with the Illumina TruSeq Nano DNA library kit (Illumina, San Diego, CA, USA) and captured using IDT xGen Exome v2 panel (Integrated DNA Technologies, Coralville, IA, USA), targeting ~34 Mb of coding region. Libraries were sequenced on a NextSeq 550 platform (Illumina, San Diego, CA, USA) to generate paired-end reads (2 × 75), and reads were aligned to GRCh37 using BWA enrichment v2.1.3.0 on BaseSpace Sequence Hub (Illumina). Variant annotation utilized Alamut Batch v1.10 (Sophia Genetics, Rolle, Switzerland), and copy number variations (CNV) in target genes (Appendix A) were identified from coverage data using Sequence Pilot v5.0 (JSI Medical Systems GmbH, Ettenheim, Germany). Sequencing quality metrics confirmed high data integrity, with an average read depth of approximately 100× across targeted exonic regions and over 95% of target bases covered at ≥20×. The average Phred quality score (Q30) exceeded 90%, and coverage uniformity was consistent with standard benchmarks for clinical-grade exome sequencing. Variants with allele frequencies > 1% (heterozygous) or >0.01% (homozygous) in gnomAD v.2.1.1 (https://gnomad.broadinstitute.org/, accessed on 30 November 2024) were excluded from variant analysis. Remaining variants were step-wise filtered and prioritized based on Clinvar entries, entries in HGMD professional, in silico predictions (Polyphen2, SIFT, DANN, MutationTaster) and were classified in accordance with the American College of Medical Genetics and Genomics (ACMG) guidelines [35].

### 2.4. Segregation

Segregation analysis was conducted using Sanger sequencing, following the protocol outlined by Haug et al. (2020) [36]. Briefly, the target region was amplified through PCR, and sequencing was carried out using BigDye™ Terminator V1.1 (Thermo Fisher Scientific, Waltham, MA, USA). The PCR products were purified on Sephadex columns and analyzed on a SeqStudio capillary sequencer (Thermo Fisher Scientific, Waltham, MA, USA). Segregation analysis was performed for all genetically solved index subjects, using DNA from the index patient and both parents if available.

## 3. Results

The study cohort comprised 24 individuals, 11 males (46%) and 13 females (54%), with a mean age of 12.4 years (1.8 to 29.0 years, SD 8 years). These participants were derived from 20 unrelated families with no reported history of consanguinity. Genetic analyses identified causative variants in 10 of the 20 families. A total of 10 potentially pathogenic variants were identified in our cohort (Table 1) including variants in crystalline proteins (*CRYBA1*, *CRYGC*, *CRYAA*), connexin proteins (*GJA8*), cytoskeletal proteins (*MIP*), membrane-associated signaling proteins (*EPHA2*), transcription factors (*MAF*), and proteins of unknown exact function (*NHS*).

Most of the variants identified were heterozygous, apart from one hemizygous variant in the *NHS* gene. The inheritance patterns in f1-8 were autosomal dominant, while two families exhibited X-linked dominant inheritance (f9 and f10) for *NHS*. Variants were primarily located in exons, except for one case involving an intronic variant in *EPHA2* (f3) which was previously reported by Zhang et al. (2009) [37]. Segregation analysis revealed maternal inheritance in most cases (f1–4, f9, f10), with affected maternal family members, while de novo variants were identified in three families (f5, f6, f8). A single instance of paternal inheritance was observed in f7.

**Table 1 biomedicines-13-01883-t001:** Identified variants through WES analysis: Functional predictions, segregation, and first reports.

Subjects		Gene	cDNA	Predicted AA Change	Ex/Int	Zygosity	Inheritance	GnomAD	ACMG	PolyPhen2	Segregation	First Publication
f1:	2-2♂ 2-3♂	*CRYBA1*	NM_005208.5: c.272_274del	p.Gly91del	ex4	het	ad	N/A	P	N/A	Maternal (affected)	Qi et al. (2003) [38]
f2	2-1♀ 2-2♂	*MAF*	NM_005360.5: c.880C>T	p.Arg294 Trp	ex1	het	ad	N/A	P	1.0	Maternal (affected)	Sun et al. (2014) [39]
f3	2-1♀	*EPHA2*	NM_004431.5: c.2826-9G>A	p.?	int16	het	ad	N/A	P	N/A	Maternal (affected)	Zhang et al. (2009) [37]
f4	3-1♀ 3-2♀	*GJA8*	NM_005267.5: c.592C>T	p.Arg198 Trp	ex2	het	ad	N/A	P	1.0	Maternal (affected)	Hu et al. (2010) [40]
f5	2-3♂	*CRYGC*	NM_020989.4: c.471G>A	p.Trp157Ter	ex3	het	ad	N/A	P	N/A	De novo	Guo et al. (2012) [41]
f6	2-2♂	*MIP*	NM_012064.4: c.97C>T	p.Arg33Cys	ex1	het	ad	N/A	P	0.982	De novo	Gu et al. (2007) [42]
f7	2-2♀	*GJA8*	NM_005267.5: c.584T>C	p.Phe195Ser	ex2	het	ad	N/A	LP	0.999	Paternal (affected)	This study
f8	2-1♀	*CRYAA*	NM_000394.4: c.347G>A	p.Arg116His	ex3	het	ad	N/A	P	1.0	De novo	Hansen et al. (2007) [43]
f9	4-2♂	*NHS*	NM_001291867.2:c.250_251insA	p.Pro84His fsTer99	ex1	hemi	xld	N/A	LP	N/A	Maternal (affected)	This study
f10	3-1♀ 3-2♀	*NHS*	NM_001291867.2:c.484del	p.Arg162Ala fsTer34	ex1	het	xld	N/A	LP	N/A	Maternal (affected)	This study

Subject identifiers (e.g., 2-2 and 3-1) refer to positions in the corresponding pedigree diagrams, following the standardized nomenclature of Bennett et al. (2022) [44], where the first number indicates the generation (from top to bottom) and the second number the birth order within that generation (from left to right). fx refers to the families. Abbreviations in alphabetical order: ACMG: American College of Medical Genetics; ad: autosomal dominant; ex: exon; f: family; hemi: hemizygous; het: heterozygous; int: intron; LP: likely pathogenic; N/A: not applicable; P: pathogenic; xld: X-linked dominant. Novel variants identified by this study are shaded in blue.

None of the identified variants were detected in gnomAD, providing strong evidence for their classification as disease-causing. PolyPhen2 scores (http://genetics.bwh.harvard.edu/pph2/, accessed on 21 November 2024), provide a highly damaging effect for all cases where applicable. All variants were classified as either pathogenic (f1–6, f8) or likely pathogenic (f7, f9, f10) based on the standard guidelines of interpretations according to the ACMG criteria [35]. Additionally, all identified variants were confirmed through Sanger sequencing to ensure accuracy and reliability. Thus, the classification as disease causing is supported by evidence of complete penetrance, consistent segregation patterns in affected families, and the absence of the variants in population databases.

### Genotype–Phenotype Comparison and Clinical Case Presentation

The families demonstrate significant variability in cataract morphology, co-existing conditions, and treatment outcomes or sequalae, respectively, as presented in Appendix A and the corresponding pedigrees of all families in the Appendix A. Figure 1 illustrates the pedigrees of families in which novel variants were identified during this study.

Cataract treatment was initiated early in most index patients, with the median age at surgery per eye being 1.5 months, and a mean age of 5.4 months (SD: 7.7 months) (Appendix A). Notably, one individual (3-3 in f10) did not require surgical intervention, as the nuclear cataract was not as dense and thus not visually limiting. Surgical procedures, including lentectomy with or without posterior capsulotomy, were performed in all cases requiring intervention. Most subjects remained aphakic following surgery, particularly those who underwent cataract removal before the age of 3 months, except for subject 2-2 in f7. Only 4 of the 14 subjects across the 10 families (Appendix A), predominantly those who were older at time of surgery, received primary intraocular lens (IOL) implantation (specifically 2-2, 2-3 in f1; 2-2 in f7; and 3-1 in f10). Early surgical intervention was prioritized to reduce the risks of amblyopia and permit vision development.

Postoperatively, all index patients received optimal refractive correction using contact lenses and/or bifocal spectacles for distance and near vision correction, respectively, with frequent assessments to adjust for refractive changes. Amblyopia treatment was employed using patching therapy to optimize visual outcomes. Secondary (aphakic) glaucoma was a significant postoperative complication, affecting 12 out of 26 eyes (46.2%). Families f1, f2, f4, f5, f7, and f8 required extensive glaucoma management, including the use of pressure-lowering eye drops, and/or multiple surgical revisions.

Vision outcomes varied significantly between eyes with and without secondary glaucoma. A two-tailed Welch’s *t*-test comparing logMAR VA between the two groups (n = 10 eyes with secondary glaucoma, n = 10 eyes without secondary glaucoma) showed a statistically significant difference (*p* = 0.049). Prior to testing data distribution and variance were assessed and found to meet the assumptions for parametric analysis. VA was significantly lower in eyes with secondary glaucoma compared to those without. The following individuals were excluded from the analysis: subject 2-3 from f10 (no cataract surgery performed), and subjects 2-1 from f3, 2-2 from f7, 4-2 from f9, whose VA could not be measured with Snellen due to young age and was limited to fixation and following behavior.

Cataract morphology varied across the cohort, with nuclear cataracts being the most common phenotype, observed in families f1 (*CRYBA1*; c.272_274del), f2 (*MAF*; c.880C>T), f5 (*CRYGC*; c.471G>A), f6 (*MIP*; c.97C>T), f8 (*CRYAA*; c.347G>A), and f10 (*NHS*; c.484del). Dense posterior cataracts were documented in f3 (*EPHA2*; c.2826-9G>A) and in 4-2 of f9 (*NHS*; c.250_252insA). Additional anterior polar cataract was found in 2-2 from f2. Total cataracts were observed in f4 (*GJA8*; c.592C>T), and cortico-nuclear cataracts in f7 (*GJA8*; c.584T>C).

Associated ocular anomalies included microphthalmia and myopia in f2, microcornea in f9, and posterior synechiae with iris atrophy in f8. Systemic comorbidities included preterm birth with perinatal asphyxia in f1 and f9, mild muscular hypotonia in f2, and cardiac anomalies such as atrial septal defect (ASD) and ventricular septal defect (VSD) in f3, all of which were not conclusively linked to the identified gene variants upon literature review.

F1 (*CRYBA1*; c.272_274del): the index subject (2-3), exhibited nuclear cataracts and underwent bilateral lentectomy at 10 months of age. His twin brother (2-2) had a similar cataract phenotype but underwent lentectomy at 1 year and 9 months. One twin (2-3) developed secondary glaucoma. Their mother (1-2) displayed nuclear cataracts, had undergone lentectomy in childhood, and achieved better vision outcomes. The father (1-1) is unaffected.

F2 (*MAF*; c.880C>T): the index subject (2-1) presented with nuclear cataracts, while her brother (2-2) exhibited both nuclear and anterior polar cataracts. Both siblings showed signs of microphthalmia at 1 month of age: 2-1 had a corneal diameter of 8.5 mm in the right eye and 9 mm in the left, with an axial length of 17.8 mm in both eyes. 2-2 displayed a corneal diameter of 9 mm and an axial length of 17.5 mm in both eyes. At 1 month of age, both individuals exhibited moderate to high myopia attributed to corneal steepening. The spherical equivalents (SE) at that age were −15.00 D (right eye) and −23.00 D (left eye) for 2-1, and −2.00 D (right eye) and −1.00 D (left eye) for 2-2. 2-1 underwent lentectomy at 1 month of age in the left and 3 months of age in the right eye and developed secondary glaucoma at 3 months in the right and at 4 years of age in the left eye. Her brother (2-2) underwent lentectomy at 1.5 months of age and achieved a better VA. Mild muscular hypotonia was observed in both siblings. The affected mother (1-2) also underwent cataract surgery due to a similar lens phenotype at a young age (exact age at surgery not reported) and displayed moderate myopia at birth, whether microphthalmia was existent at birth was not documented.

F3 (*EPHA2*; c.2826-9G>A): the index subject (2-1) presented with very dense posterior cataracts. Lentectomy was performed at 1.5 months of age. Her father (1-2) also had CC and underwent lentectomy at 6 years of age. The congenital cardiac anomalies in 2-1, are not reported to be associated with the variant.

F4 (*GJA8*; c.592C>T): the index (2-2), and her sister (2-1), had total cataracts requiring early surgical intervention. Both siblings developed secondary glaucoma. Their mother (1-2) also had total cataracts, her surgical intervention was not documented.

F5 (*CRYGC*; c.471G>A): index subject (2-3), had nuclear cataracts and underwent surgery at 1 month. He is the only carrier of the variant thus it being de novo.

F6 (*MIP*; c.97C>T): subject 2-2 presented with bilateral nuclear cataracts. Both parents are unaffected. He underwent surgery at 1.5 months of age. Vision outcomes were favorable.

F7 (*GJA8*; c.584T>C): 2-2, the index in f7, displayed cortico-nuclear cataracts and received cataract surgery at 1.5 months of age. Aphakic glaucoma developed postoperatively requiring multiple interventions. Her affected father (1-1) underwent surgery in infancy without documented glaucoma development.

F8 (*CRYAA*; c.347G>A): the only affected subject is index 2-1. She displayed bilateral nuclear cataracts and underwent lentectomy at 1.5 months in the left 2 months in the right eye. Postoperative complications included aphakic glaucoma, posterior synechiae, and nystagmus, which significantly affected vision outcomes.

F9 (*NHS*; c.250_252insA): CC and teeth anomalies were presented in f9. The latter noted only in the mother (3-2) of the index (4-2). Cataract phenotypes varied among family members: the grandmother (2-2) exhibited nuclear cataracts meanwhile her sister (2-3) did not display CC, the mother’s (3-2) cataract phenotype was not documented, and the index subject (4-2) had a very dense posterior cataract. The sister of the grandmother (2-3) did not undergo cataract surgery while surgical interventions were performed in 3-2 at 3 months, in 2-2 at 61 years, and in 1-2 at 30 years of age. Additionally, the index (4-2) displayed microcornea. No affected family member developed secondary glaucoma.

F10 (*NHS*; c.484del): in most affected family members a nuclear type of cataract was described. The index subject (3-2) had the mildest form and does not require surgery to date. Her sister (3-1) exhibited denser visually impairing nuclear cataracts, which required surgery at 1 year and 9 months on the right and 2 years and 3 months in the left eye. Their mother (2-2) had undergone surgery in her third decade of life. The maternal aunt (2-3) carries the variant but remained unaffected. The grandmother (1-2) underwent bilateral cataract surgery at 30 years of age. None of the affected family members developed secondary glaucoma. The affected family member did not display non-ocular manifestations of NHS.

## 4. Discussion

The findings in this study expand the current understanding of the genetic basis of CCs by identifying two novel variants in *NHS* (c.250_252insA and c.484del) and one in *GJA8* (c.584T>C). These variants not only broaden the mutational spectrum of CC-associated genes but also underscore the importance of genetic profiling for the accurate diagnosis of syndromic and non-syndromic CC cases.

A resent USA-based study by Rossen et al. (2023) similarly applied WES in a diverse pediatric cohort with CCs, identifying causative variants in 46.2% cases [45]. In comparison, our Swiss cohort yielded a similarly high diagnostic rate of 50%, supporting the robustness of WES across different population and study settings.

The identification of these variants further highlights the phenotypic variability in *NHS*-associated CCs, offering insights into the roles of X-chromosome inactivation, dose dependency, and mosaicism in modulating disease severity. The *NHS* gene encodes the NHS actin remodeling regulator, a protein essential for the development of eyes, teeth, craniofacial structures, and brain. It plays a pivotal role in maintaining cell morphology by regulating actin dynamics, which is crucial for cytoskeletal integrity [46]. Located at Xp22.13, *NHS* comprises 10 coding exons, undergoes alternative splicing, and produces multiple transcript variants [46,47,48].

Variants in *NHS* lead to both NHS and isolated X-linked CCs, though clear genotype–phenotype correlations remain elusive. Complete loss of NHS protein typically leads to the full NHS phenotype, while disruptions in gene regulation may result in milder, non-syndromic CCs [30,49]. Previous database analyses have identified over 100 disease-associated *NHS* variants, with point mutations and small indels linked to syndromic NHS, while larger structural variations were found to be associated with X-linked cataracts [30,46,50,51,52].

These findings are consistent with studies in other populations. For instance, Li et al. (2018) [50] reported a Chinese pedigree with a frameshift *NHS* variant showing full NHS phenotype in affected males and variable lens opacities in heterozygous females. Similarly, Sun et al. (2014) [39] identified *NHS* variants among Chinese families with non-syndromic CCs, highlighting incomplete penetrance. These phenotypic patterns closely mirror our Swiss cohort, where affected females exhibited a broad range of presentations despite carrying the same variant. Although we did not systematically assess ethnicity, the similarity of our findings to reports from Asian, European, and North American cohorts highlights the global relevance of both *NHS* and *GJA8* variants in the etiology of CCs [39,43,53,54,55,56]. In our cohort, two novel *NHS* frameshift indels—c.250_252insA in f9 and c.484del in f10—were identified, both showing striking intrafamilial phenotypic variability. For example, in f10, the index patient (3-2) had mild nuclear cataract without vision impairment, whereas her sister (3-1) required early lentectomy due to dense nuclear cataracts. Their mother (2-2) underwent cataract surgery in adulthood, while the aunt (2-3), carrying the same variant, exhibited no lens opacities at all. This pattern illustrates the role of X-chromosome inactivation and mosaicism across affected females [16,30].

Heterozygous females are somatic mosaics for *NHS* expression—if the X chromosome carrying the wild-type allele is preferentially inactivated, the mutant allele is expressed in more cells, leading to a more severe phenotype [57]. Conversely, skewed inactivation of the mutant X can result in little or no disease manifestation [49]. This mechanism explains how some carriers remain asymptomatic while others require early surgical intervention, as observed in f9 and f10, where some individuals present with syndromic features while others did not.

These observations support a dose-dependent model for *NHS*-related disease, in which lower levels of functional NHS protein is associated with more severe cataracts and, in some cases, additional syndromic features [58]. Studies on X-inactivation in female carriers have revealed that some display symptoms as severe as hemizygous males, reinforcing the importance of X-linked dosage effects [18]. Coccia et al. (2009) [49] further demonstrated that NHS and X-linked CCs are allelic disorders caused by different mutations in the same gene. While complete loss-of-function mutations typically result in syndromic NHS, variants affecting gene regulation may cause isolated cataract. However, our findings suggest that this distinction alone does not fully explain the observed variability. Additional influences, such as skewed X-inactivation and the presence of genetic modifiers, may also influence the clinical presentation in both female carriers and some affected males [21]. To date, no specific modifier genes have been definitively identified for NHS; however, several candidates involved in X-inactivation, chromatin remodeling, cytoskeletal regulation, and lens development are under investigation based on their biological relevance and overlapping expression patterns [21,49,59]. Thus, while modifiers are likely involved, their identity and mechanisms remain to be elucidated.

The novel *GJA8* (c.584T>C) variant in f7 also contributes significantly to CC understanding. The *GJA8* gene encodes connexin 50, a transmembrane protein that forms gap junction channels essential for lens growth, fiber cell maturation, and maintaining transparency [60]. *GJA8* variants have been widely linked to autosomal dominant CCs, often causing disrupted gap junction communication, protein trafficking, or hemichannel function [61]. According to the HGMD there are 113 known disease-associated *GJA8* variants thus far (https://www.hgmd.cf.ac.uk, accessed on 6 May 2025). The newly identified *GJA8* c.584T>C variant was associated with cortico-nuclear CCs, where the affected infant (2-2) required early surgical intervention. This phenotype is consistent with other reported *GJA8* variants [22,53]. For example, Merepa et al. (2024) [53] reported multiple pathogenic missense *GJA8* variants in a large European cohort, including various missense *GJA8* variants associated with similar CCs.

Our study is limited by a small cohort size, restricting the generalizability of findings. WES may have missed deep intronic or structural variants, and pathogenicity was inferred without functional validation. We also did not systematically assess the ethnicity of participants (despite the Swiss of mainly Caucasian background), which may limit conclusions about variant prevalence or expressivity across different population groups. Moreover, the primary aim of this study was not to provide generalizable epidemiological data but to contribute to the expanding catalog of CC-associated variants. These limitations may lead to an underestimation of the true genetic burden and reduce the ability to draw firm conclusions about rare genotype–phenotype correlations or variant-specific effects. In particular, the small sample size limits statistical power and may not capture the full phenotypic spectrum seen in larger, more diverse populations. Despite these limitations, the observed genotype–phenotype patterns align with previous studies across diverse populations and expand the known variant spectrum.

These findings support routine genetic testing in CC, particularly in familial cases. Early molecular diagnosis can guide clinical decisions, enable targeted counseling, and prompt family screening. Future studies should aim to refine genotype–phenotype correlations and identify factors that modulate disease expression.

## 5. Conclusions

This study expands the genetic understanding of CCs by identifying novel DNA sequence variants and highlighting the genetic heterogeneity present in a Swiss cohort. Through WES, we identified three previously unreported newly discovered variants—*GJA8* (c.584T>C), *NHS* (c.250_252insA), and *NHS* (c.484del)—providing new insights into the molecular mechanisms of CCs. Additionally, previously documented variants in *CRYBA1*, *CRYGC*, *CRYAA*, *MIP*, *EPHA2*, and *MAF* confirm the complex genetic presentation of CCs in a Swiss cohort.

## Figures and Tables

**Figure 1 biomedicines-13-01883-f001:**
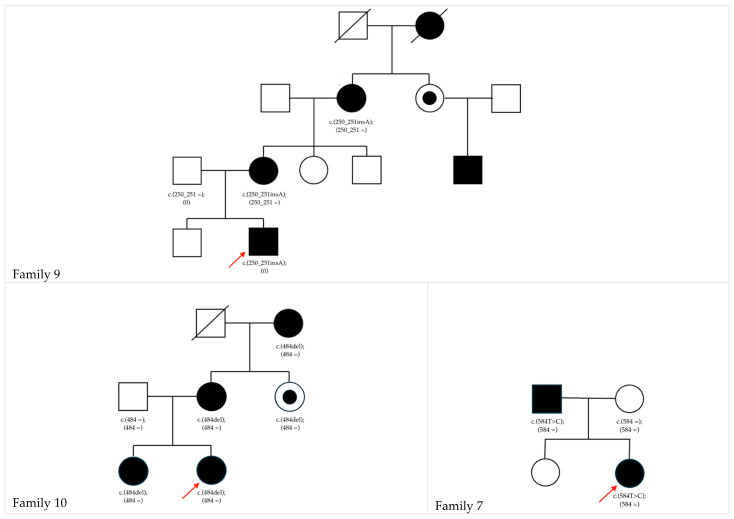
Pedigrees of families with novel gene variants found in this study. Pedigrees showing segregation of the c.250_251insA variant in *NHS* in family 9, the c.484del variant in *NHS* of family 10, and the c.584T>C variant in *GJA8* in family 7. The red arrow indicates the index patient of the family. Squares represent males, circles represent females. Filled symbols indicate affected individuals, half-filled symbols indicate carriers, and open symbols denote unaffected individuals. A diagonal line through a symbol indicates that the individual is deceased. Generations are numbered from top to bottom, with the top row representing the first generation. Within each generation, individuals are arranged from left to right in birth order, with the leftmost symbol representing the first-born [44].

## Data Availability

The original contributions presented in this study are included in the article/Appendix A. Further inquiries can be directed to the corresponding authors.

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
