# Peer review of "Genetic Landscape of Congenital Cataracts in a Swiss Cohort: Addressing Diagnostic Oversights in Nance–Horan Syndrome"

_biomedicines, 2025, doi:10.3390/biomedicines13081883_

Round 1
Reviewer 1 Report
Comments and Suggestions for Authors
In the manuscript, the authors sequenced 20 congenital cataracts patients. The whole-exome sequencing (WES) was conducted based on the list of genes previously compiled and augmented. Combining with the family data, they identified. They identified 10 new mutations associated with CC. Among those, two additional NHS variants was also identified. This study tackles one of the understudied and underdiagnosed disease. The identification of new variants will provide new insights in diagnosing CC. One small detail that needs to be added to the manuscript is the quality of the sequencing, depth, and coverage of target sequences.
Reviewer 2 Report
Comments and Suggestions for Authors
The work by Delas et al., provides a valuable and comprehensive contribution to the genetic understanding of congenital cataracts, particularly by identifying novel NHS and GJA8 variants and highlighting the diagnostic challenges of syndromes like Nance-Horan Syndrome. The integration of detailed clinical and genetic data, clear methodology, and thoughtful discussion of genotype-phenotype variability makes this work highly relevant for both clinical practice and future research; addressing some points related to data presentation, clinical implications, and limitations will further enhance its clarity and impact.
Introduction
- Lines 43/44: “the NHS gene is the only known X-linked dominant (xld) gene associated with CC", please clarify whether other X-linked genes (e.g., OCRL in Lowe syndrome) are excluded or if NHS is uniquely xld.
- Line 73: "Cat-Map database has cataloged over 450 genes." Please briefly describe Cat-Map’s purpose (e.g., "a curated resource for cataract-associated loci") for non-specialist readers.
- Lines 86/87: "102 disease-causing variants...HGMD" the authors are recommended to compare this to other CC-related genes (e.g., "NHS variants represent ~0.2% of all CC-associated mutations") to underscore rarity.
- Line 105: “NHS variants demonstrated phenotypic variability suggestive of dose dependency.” The authors should define "dose dependency" in the context of X-linked dominant inheritance and provide an example from the literature.
Methods
- Section 2.1.:
a- The total number of subjects and families included is not stated in this section. The authors should include a sentence specifying the cohort size and any key inclusion or exclusion criteria.
b- While comprehensive exams are mentioned, the specific clinical criteria or grading systems for cataract morphology, severity, and associated anomalies are not detailed. Please briefly describe the clinical assessment protocols or reference standardized grading systems used.
C- Also, there is no mention of how missing data, ambiguous clinical findings, or sample quality issues were handled.
- Section 2.3.: Exome sequencing
a- There is no information on average sequencing depth, coverage statistics, or quality thresholds for variant calling.
b- Although allele frequency thresholds are given, the criteria for pathogenicity classification (e.g., ACMG guidelines, in silico prediction tools, literature review) are not described.
- Segregation subtitle section should be numbered 2.4.
- It is not specified whether segregation analysis was performed for all candidate variants or only a subset, nor how incomplete family data were handled.
Results
- Line 211: A t-test is mentioned regarding vision outcomes with and without secondary glaucoma. Please provide the exact p-value, sample sizes, and clarify if the data met the assumptions for parametric testing.
- The authors may consider including summary statistics (e.g., mean/median age at surgery, complication rates) in a concise table for the entire cohort.
Discussion
- Lines 326–336: While the discussion references several studies, it could be more explicitly compared to the findings of this study with previous reports of NHS and GJA8 variants in similar populations or cohorts. For example, how do the observed phenotypes in this Swiss cohort compare to those reported in other ethnic or geographic groups?
- Lines 335–340: The mention of potential genetic modifiers is important but remains speculative. If possible, briefly discuss any specific candidate genes or pathways under investigation, or reference recent reviews on this topic.
- Future perspectives and clinical implications should be addressed by the end of the discussion. The discussion could be strengthened by explicitly stating how the current findings should inform clinical practice, such as providing recommendations for genetic testing in CC, emphasizing the importance of family screening, or outlining implications for early intervention and counseling.
- Lines 351–354: While limitations are acknowledged, the discussion could more clearly address how these limitations might affect the interpretation or generalizability of the results.
Reviewer 3 Report
Comments and Suggestions for Authors
In this manuscript the authors explore genetic makeup of 20 unrelated Swiss families with congenital cataracts by whole-exome sequencing (WES). Particularly, focused on Nance-Horan Syndrome (NHS) because frequently is underdiagnosed principally in female carriers. Reported on three novel variants: one in GJA8 gene and two NHS gene. The results obtained provide new insights into the genetic complexity of variants in congenital cataracts.
In general, the introduction supported the objective proposed, the methods used were adequate, the description of results and the discussion section was acceptable. The conclusion is supported by the description of results. I think the manuscript is promising, I would suggest the authors considerate change the nomenclature in Pedigrees to a new Standardized nomenclature propose by National Society of Genetic Counselors. (Bennett RL, et. al. J Genet Couns. 2022 31(6):1238-1248. doi: 10.1002/jgc4.1621).
